# Flower-Shaped Plasma Cells in Multiple Myeloma with Morphological Heterogeneity

**DOI:** 10.3390/diagnostics14202285

**Published:** 2024-10-14

**Authors:** Hiroki Hosoi, Misato Tane, Makiko Sogabe, Ryuta Iwamoto, Naoto Minoura, Shogo Murata, Toshiki Mushino, Akinori Nishikawa, Shin-Ichi Murata, Takashi Sonoki

**Affiliations:** 1Department of Hematology/Oncology, Wakayama Medical University, Wakayama 641-8509, Japanshogo@wakayama-med.ac.jp (S.M.); nishikaw@wakayama-med.ac.jp (A.N.);; 2Department of Transfusion Medicine, Wakayama Medical University Hospital, Wakayama 641-8510, Japan; 3Department of Human Pathology, Wakayama Medical University, Wakayama 641-8509, Japan; riwamoto@wakayama-med.ac.jp (R.I.);; 4Department of Clinical Laboratory, Wakayama Medical University Hospital, Wakayama 641-8510, Japan; 5Division of Medical Information, Wakayama Medical University Hospital, Wakayama 641-8510, Japan

**Keywords:** flower-shaped nuclei, multiple myeloma, morphological heterogeneity

## Abstract

Background: Flower-shaped nuclei in plasma cells are rare in multiple myeloma. Case presentation: We report on an 88-year-old male who presented with a mass lesion in the clavicular region. A biopsy of the mass revealed an increase in mature plasma cells with round nuclei. In contrast, a bone marrow examination showed increased plasma cells with flower-shaped nuclei. The patient tested negative for human T-lymphotropic virus type-1 and was diagnosed with multiple myeloma. Conclusions: While multiple myeloma is known for intra-tumor heterogeneity, reports of morphological heterogeneity based on the site of tumor sampling are limited. In this case, the presence of plasma cells with flower-shaped nuclei enabled the identification of site-dependent morphological tumor heterogeneity.

Multiple myeloma (MM) is a plasma cell neoplasm, and one of the diagnostic criteria is the clonal proliferation of plasma cells in the bone marrow [1]. Although the tumor cells of MM resemble normal plasma cells in most cases, unusual morphological variants are sometimes observed in MM, including small, anaplastic, and plasmablastic cells [2]. Multilobulated cells were also reported as flower-shaped plasma cells mimicking the flower cells of adult T-cell leukemia/lymphoma (ATLL) [3,4]. However, the characteristics of MM exhibiting a flower-shaped morphology remain unclear due to the rarity of this subtype. We report an MM patient with flower-shaped plasma cells with morphological heterogeneity depending on the site of tumor involvement.

An 88-year-old male presented with a tumor in the left clavicular region. He had suffered a left clavicle fracture one year earlier. A radiographic examination revealed a lytic bone lesion (Figure 1a). The gadolinium-enhanced magnetic resonance imaging of the left clavicular area revealed a 55 mm soft tissue mass (Figure 1b,c). A tumor biopsy of the left clavicular region revealed clusters of plasma cells, leading to the diagnosis of a plasma cell neoplasm. The majority of the plasma cells had round nuclei (Figure 2a). Immunohistochemical staining showed that the plasma cells were positive for CD138, CD56, and cyclin D1 (Figure 2b,c). In situ hybridization for immunoglobulin light chains showed lambda positivity, indicating a skewed expression of light chains (Figure 2d,e). No anemia, renal insufficiency, or hypercalcemia developed. Bone involvement was also observed in the sternum and ribs. The serum immunoglobulin G (IgG) level was elevated to 2727 mg/dL, and the IgG-λ monoclonal protein was detected. The serum free light-chain assay showed serum kappa—19.4 mg/L (reference range: 3.3–19.4), serum lambda—123.8 mg/L (reference range: 5.7–26.3), and a decreased kappa/lambda ratio of 0.16 (reference range: 0.26–1.65).

Bone marrow (BM) aspiration showed 43% of the abnormal cells with flower-shaped nuclei, morphologically distinct from the biopsy of the clavicular mass (Figure 3a,b). The abnormal cells observed in the bone marrow had a morphology typical of ATLL. However, a test for human T-lymphotropic virus type-1 (HTVL-1) antibodies was negative, ruling out a diagnosis of ATLL. The flow cytometry of the BM aspiration sample showed that the abnormal cells were positive for CD38, CD56, CD138, and cytoplasmic λ and negative for CD7, CD19, and CD20 (Figure 3c,d). The results indicated that the abnormal cells were of plasma cell origin. The cytogenetic analysis using fluorescence in situ hybridization showed that they were negative for *TP53* deletion, 1q21 amplification, and *FGFR3* and *MAF* translocations. The G-banding analysis revealed a normal karyotype. The patient was diagnosed with MM. The mass lesion in the left clavicular region was diagnosed as an extramedullary plasmacytoma of MM. He received a combination of bortezomib and dexamethasone considering his age and frailty. After two weekly doses of bortezomib, the mass in the left clavicular region had reduced in size, and the patient was transferred to a hospital near his home.

Although flower-shaped plasma cells are rare, several cases have been reported [3,4,5,6,7,8,9,10,11]. No studies have investigated the morphological differences across multiple sites within the same patient presenting flower-shaped plasma cells. MM is known to exhibit intra-patient and intra-tumor heterogeneity; however, few reports describe morphological diversity based on the sites of tumor involvement [8,12]. In our patient, a distinctive flower-shaped morphology was observed in the bone marrow, enabling the identification of morphological differences depending on the site of tumor involvement. Our case suggests that the morphology of plasma cells may vary according to the lesion site. On the other hand, it should be noted that the differences in the specimen-collection methods between biopsy and aspiration may also influence its morphological findings. Further investigations involving biopsies from multiple sites are warranted to explore the morphological heterogeneity of MM showing flower-shaped nuclei.

## Figures and Tables

**Figure 1 diagnostics-14-02285-f001:**
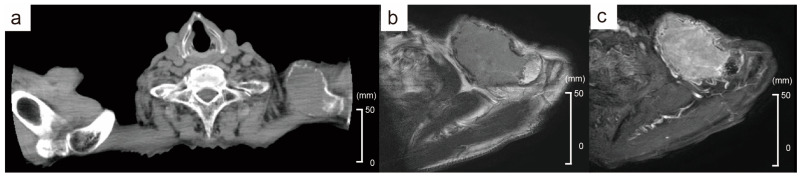
Radiological findings of the left clavicular mass. (**a**) CT finding shows a lytic mass. (**b**) MRI T2-weighted image reveals a 55 mm soft tissue mass. (**c**) Gadolinium-enhanced MRI. The red arrows indicate the tumor.

**Figure 2 diagnostics-14-02285-f002:**
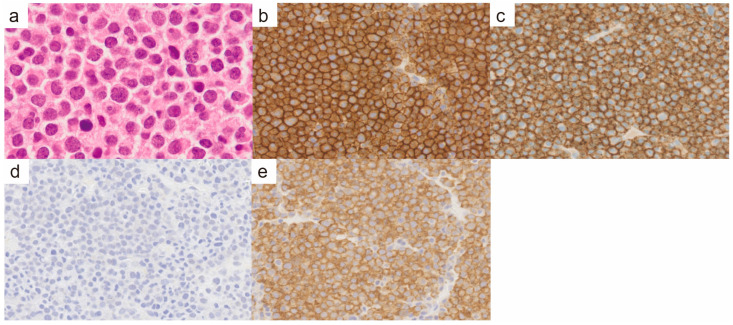
Pathological findings of the left clavicular mass. (**a**) Hematoxylin and eosin staining (×400). There were very few plasma cells with cleaved nuclei, and the majority of plasma cells had round nuclei. (**b**) CD138 immunohistochemistry (×400). (**c**) CD56 immunohistochemistry (×400). (**d**) In situ hybridization for immunoglobulin light chain kappa (×400). (**e**) In situ hybridization for immunoglobulin light chain lambda (×400).

**Figure 3 diagnostics-14-02285-f003:**
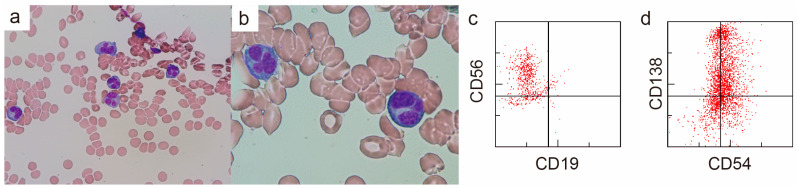
Findings of bone marrow aspiration from the left iliac bone (**a**,**b**) Wright–Giemsa stain, (**a**) ×400, (**b**) ×1000. The majority of plasma cells had flower-shaped nuclei. (**c**,**d**) Surface antigen analysis by flow cytometry.

## Data Availability

The data presented in this study are available from the corresponding author upon reasonable request.

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
