# Peer review of "Flower-Shaped Plasma Cells in Multiple Myeloma with Morphological Heterogeneity"

_diagnostics, 2024, doi:10.3390/diagnostics14202285_

Round 1
Reviewer 1 Report
Comments and Suggestions for Authors
It is a very rare interesting case of flower-shaped plasma cells in multiple myeloma.
My notes:
1. Please indicate a size of tumor in clavicula region if possible (was it extramedullary plasmacytoma?)
2. Is it possible to include some figures with immunohistochemical staining? It will be great.
3. Please, add information about serum free light chain ratio.
4. What about CD19 expression in abnormal plasma cells in this case? Please, indicate it in the description of immonophenotype.
5. It will be great if authors add some images of flow cytometry results. But it up to authors.
Author Response
Point-by-point replies to the comments from reviewer #1
Reviewer’s comment
It is a very rare interesting case of flower-shaped plasma cells in multiple myeloma.
Our response
Thank you for your favorable comments. Our point-by-point responses are shown below.
Comment 1
Please indicate a size of tumor in clavicula region if possible (was it extramedullary plasmacytoma?)
Our response
Thank you for your valuable comment. We have included the size of the tumor in the clavicular region (lines 35–36). Additionally, we have updated the diagnosis to indicate that this was an extramedullary plasmacytoma (lines 73–74). As you pointed out, the information about the tumor is important, and we have included an image of the tumor lesion in the Figure (Figures 1a–1c).
Comment 2
Is it possible to include some figures with immunohistochemical staining? It will be great.
Our response
We have added figures with immunohistochemical staining (Figures 2b–2e).
Comment 3
Please, add information about serum free light chain ratio.
Our response
As per your suggestion, we have added the serum free light chain test results. (lines 44–47)
Comment 4
What about CD19 expression in abnormal plasma cells in this case? Please, indicate it in the description of immonophenotype.
Our response
CD19 was negative, and we have added this information regarding CD19 expression in the description of the immunophenotype. (line 69)
Comment 5
It will be great if authors add some images of flow cytometry results. But it up to authors.
Our response
Thank you for your suggestion. We have added the flow cytometry results to Figures 3c and 3d.
Reviewer 2 Report
Comments and Suggestions for Authors
This is a nice representation of an unusual morphology of plasma cells. However, the following comments would need to be address by the authors:
1. The images need to be modified:
Present two panels:
First panel: Clavicle: H&E, CD138
Second panel: Bone marrow: H&E core biopsy, aspirate smear, CD138
In order to distinctly depict the morphology difference in clavicle and bone marrow, please present a H&E image of the bone marrow core biopsy. While it is possible the flower shaped cells are present on the aspirate, it is necessary to see if such a morphology difference could be observed on the HE BM core biopsy. On the other hand, clavicle mass don't get a touch prep and therefore the extent of flower shaped plasma cells could not be reliably assessed. The H&E from the clavicle does appear to show occasional cleaved/flower shaped nuclei in the cells of interest.
2. Flow cytometry findings: please add the negative T-cell markers in addition to CD7
3. The authors mention plasma cells are positive for cyclinD1: this is more frequent in t(11;14) plasma cell myeloma. Was FISH analysis attempted to detetect this? Are there any positive FISH results as it pertains to Myeloma.
4. Please modify this sentence 'The treatment improved the clavicular mass' with appropriate radiologic/clinical response criteria
Author Response
Point-by-point replies to the comments from reviewer #2
Reviewer’s comment
This is a nice representation of an unusual morphology of plasma cells. However, the following comments would need to be address by the authors.
Our response
Thank you for your favorable comments. Our point-by-point responses are shown below.
Comment 1
The images need to be modified:
Present two panels:
First panel: Clavicle: H&E, CD138
Second panel: Bone marrow: H&E core biopsy, aspirate smear, CD138
In order to distinctly depict the morphology difference in clavicle and bone marrow, please present a H&E image of the bone marrow core biopsy. While it is possible the flower shaped cells are present on the aspirate, it is necessary to see if such a morphology difference could be observed on the HE BM core biopsy. On the other hand, clavicle mass don't get a touch prep and therefore the extent of flower shaped plasma cells could not be reliably assessed. The H&E from the clavicle does appear to show occasional cleaved/flower shaped nuclei in the cells of interest.
Our response
As you suggested, we have divided the histological findings into two separate figures.
Thank you for your critical comment. As you correctly pointed out, comparing the HE-stained specimens of the bone marrow biopsy and the clavicular tumor biopsy would be ideal. Unfortunately, in this case, only a bone marrow aspiration was performed, and neither a bone marrow biopsy nor a bone marrow clot specimen was evaluated. Therefore, we are unable to present the results of a bone marrow biopsy. This is one of the limitations of this case report, and we have mentioned it in the discussion section. (lines 91–93)
Comment 2
Flow cytometry findings: please add the negative T-cell markers in addition to CD7.
Our response
Thank you for your comment. Unfortunately, in this case, we did not assess the expression of T-cell markers other than CD7.
Comment 3
The authors mention plasma cells are positive for cyclinD1: this is more frequent in t(11;14) plasma cell myeloma. Was FISH analysis attempted to detetect this? Are there any positive FISH results as it pertains to Myeloma.
Our response
Thank you for your important comment. While the plasma cells were positive for cyclin D1, we agree that t(11;14) should have been confirmed by FISH analysis. Unfortunately, this test was not performed in this case. At our institution, FISH analysis is typically performed only for high-risk cytogenetic markers in multiple myeloma cases. The results of the FISH tests that were conducted are included in the manuscript. (lines 70–71)
Comment 4
Please modify this sentence 'The treatment improved the clavicular mass' with appropriate radiologic/clinical response criteria.
Our response
Thank you for pointing out the non-specialized terminology. As you suggested, the description should follow appropriate radiologic and clinical response criteria. However, in this case, the patient was transferred to a long-term care hospital near his home shortly after initiating treatment, so we were unable to assess the treatment response at our institution. We have revised the sentence to reflect this situation accurately (lines 75–76). Since this case report focuses on the diagnosis, we do not have follow-up data on the patient’s post-treatment course.